# Recapitulation of pathophysiological features of AD in SARS-CoV-2-infected subjects

Elizabeth Griggs[1], Kyle Trageser[1], Sean Naughton[1], Eun-Jeong Yang[1], Brian Mathew[1], Grace Van Hyfte[1], Linh Hellmers[2], Nathalie Jette[1], Molly Estill[1], Li Shen[1], Tracy Fischer[2,3], Giulio Maria Pasinetti[1,4]*

[1]Department of Neurology, Icahn School of Medicine at Mount Sinai, New York, United States; [2]Tulane National Primate Research Center, Covington, United States; [3]Department of Microbiology and Immunology, Tulane University School of Medicine, New Orleans, United States; [4]Geriatric Research, Education and Clinical Center, James J. Peters Veterans Affairs Medical Center, New York, United States

**Abstract** Infection with the etiological agent of COVID-19, SARS-CoV-2, appears capable of impacting cognition in some patients with post-acute sequelae of SARS-CoV-2 (PASC). To evaluate neuropathophysiological consequences of SARS-CoV-2 infection, we examine transcriptional and cellular signatures in the Brodmann area 9 (BA9) of the frontal cortex and the hippocampal formation (HF) in SARS-CoV-2, Alzheimer's disease (AD), and SARS-CoV-2-infected AD individuals compared to age- and gender-matched neurological cases. Here, we show similar alterations of neuroinflammation and blood–brain barrier integrity in SARS-CoV-2, AD, and SARS-CoV-2-infected AD individuals. Distribution of microglial changes reflected by the increase in Iba-1 reveals nodular morphological alterations in SARS-CoV-2-infected AD individuals. Similarly, HIF-1α is significantly upregulated in the context of SARS-CoV-2 infection in the same brain regions regardless of AD status. The finding may help in informing decision-making regarding therapeutic treatments in patients with neuro-PASC, especially those at increased risk of developing AD.

*For correspondence:
giulio.pasinetti@mssm.edu

## Editor's evaluation

This study compares neuropathological changes in postmortem brain samples from patients with SARS-CoV-2 infection, Alzheimer's disease, both SARS-CoV-2 and Alzheimer's disease, and demographically matched controls. This is an important comparison that expands information on SARS-CoV-2 effects in the brain. Solid results are presented to support the main claims.

## Introduction

The consequences of SARS-CoV-2 infection have been well studied in the respiratory system; however, much less information is available regarding the neurological consequences of infection (*Flerlage et al., 2021*). Previous evidence tentatively suggests that SARS-CoV-2 may be neuroinvasive, leading to a vast array of neurological symptoms, including anosmia, ageusia, cognitive functions, and cerebrovascular disorders (*Desforges et al., 2014*; *Hingorani et al., 2022*). Neurological complications of SARS-CoV-2 infection manifest with increased severity related to age and shared medical history of metabolic disorders and other vascular risk factors (*Divani et al., 2020*; *Chou et al., 2021*; *Sullivan and Fischer, 2021*). Some strains within the coronavirus family are implicated in neuronal degeneration, such as HCoV-OC43, which can lead to glutamate excitotoxicity and neuronal degradation in

mice through cytokine production (*Jacomy et al., 2010*; *Brison et al., 2011*). Because SARS-CoV-2 is a new addition to the coronavirus family, less information exists regarding the potential long-term neurological complications that may impact COVID-19 survivors. This may be especially concerning for aging individuals with increased vulnerability to developing age-related neurodegenerative diseases.

While it is currently unknown whether neurological manifestations of SARS-CoV-2 infection arise solely from systemic inflammation in the periphery or brain infiltration, it is well established that secretion of cytokines and chemokines from the periphery allows the recruitment of leukocytes and other cells to specific tissues (*Bauer et al., 2022*; *Prinz and Priller, 2017*). This type of immune response is speculated to accelerate blood–brain barrier (BBB) disruption and may potentially damage cells within the central nervous system (CNS) (*de Erausquin et al., 2021*). Such effects are similar to neuropathology seen in Alzheimer's disease (AD) and other neurodegenerative disorders, where leukocyte infiltration, BBB dysregulation, and microglial activation are observed. Thus, it is possible that SARS-CoV-2 may accelerate the onset and severity of cognitive decline or AD in susceptible individuals (*Meinhardt et al., 2021*; *Mhatre et al., 2015*). Understanding the impact of SARS-CoV-2 infection on the CNS and subsequent mechanisms associated with cognition is particularly pressing, given the number of individuals presenting with neurological symptoms of post-acute sequelae of SARS-CoV-2 (neuro-PASC), colloquially termed 'long-COVID' (*Shanley et al., 2022*; *Davis et al., 2021*).

Here, we investigate the potential brain mechanisms associated with SARS-CoV-2 infection in SARS-CoV-2, AD, and SARS-CoV-2-infected AD individuals by comparing transcriptional and cellular responses in the cortical Brodmann area 9 (cortical BA9) of the frontal cortex and the hippocampal formation (HF); two brain regions deeply involved in cognitive and emotional functions, to age- and gender-matched neurological cases. Here, we report evidence suggesting that SARS-CoV-2 infection promotes similar pathophysiological features found in AD cortical BA9 and HF regions and possibly exacerbate preexisting AD pathophysiology.

## Results

### Demographics of human postmortem cases

We examined postmortem tissue samples of cortical BA9 of the frontal cortex and the HF collected by the Neuropathology Brain Bank and Research CoRE at Mount Sinai from four different groups:

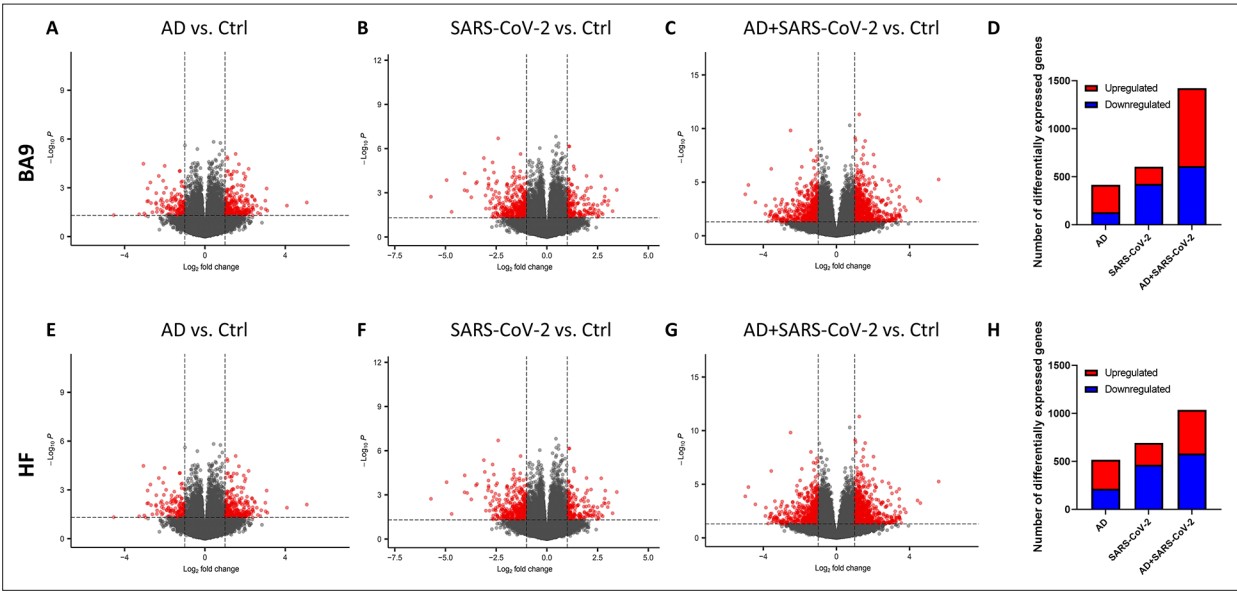

**Figure 1.** Gene expression in cortical Brodmann area 9 (BA9) and hippocampal formation (HF). (**A–C**) Volcano plot distribution of gene transcripts of the cortical BA9 region in Alzheimer's disease (AD) cases (**A**), SARS-CoV-2 cases (**B**), and SARS-CoV-2-infected AD cases (**C**) compared to neurological control cases. (**E–G**) Volcano plot distribution of gene transcripts of the cortical HF region in AD cases (**E**), SARS-CoV-2 cases (**F**), and SARS-CoV-2-infected AD cases (**G**) compared to neurological control cases. Volcano plots were generated from 39,901 genes (**A, B, E, F**), 38,021 genes (**C**), and 35,527 genes (**G**). Transcripts with nominal p<0.01 and an absolute log2 fold-change (log2 FC) > 1 are indicated in red. (**D, H**) Bar graphs, in which the number of upregulated and downregulated genes (with p<0.01 and absolute value of log2 FC > 1), is indicated in red and blue, respectively.

SARS-CoV-2, AD, and AD individuals who became infected by SARS-CoV-2, compared to age- and gender-matched neurological cases (*Supplementary file 1*). Each group comprises an equal number of age-matched male and female patients, ensuring equal distribution with an average age of 79.6 among groups. Medical records indicate that all AD individuals have postmortem ABC scores indicative of AD (*Supplementary file 2*), where A is a measure of amyloid β deposition, B is a measure of neurofibrillary degeneration based on the Braak and Brook score, and C is scored based on neuritic plaques outlined by the Consortium to Establish a Registry for Alzheimer's Disease diagnosis (CERAD) (*Kovacs and Gelpi, 2012*). SARS-CoV-2 infection was confirmed by diagnostic polymerase chain reaction (PCR). Within the SARS-CoV-2 and SARS-CoV-2-infected AD groups, all patients were symptomatic, admitted to the hospital, and received oxygen supplementation by ventilator or cannula with disease onset to death occurring on average 27 d after diagnosis for patients with SARS-CoV-2 only and 32 d after diagnosis in infected AD patients (*Supplementary files 1 and 2*). Blood specimens were collected and indicate several laboratory features of severe COVID-19 (*Supplementary file 2*), such as increased C-reactive protein (CRP) and interleukin-6 (IL-6) (*Lampart et al., 2022*).

## Gene expression of SARS-CoV-2 and AD groups compared to control cases in cortical BA9 and HF

The transcriptional profiles of AD and SARS-COV-2-infected postmortem cases from the cortical BA9 were assessed by pairwise comparison to determine gene expression compared to neurological control cases (*Figure 1A–C*). Volcano plot distributions of gene transcripts of AD and SARS-CoV-2 cases compared to neurological controls (*Figure 1A and B*) both reveal 39,901 genes that were used to determine differentially expressed genes based on a nominal p-value<0.01 and an absolute log2 fold-change (log2 FC) > 1. In the AD group, 287 genes were upregulated and 131 genes were downregulated (*Figure 1D*). In the SARS-CoV-2 group, 179 genes were upregulated and 426 genes were downregulated (*Figure 1D*). The volcano plot distribution of gene transcripts of SARS-CoV-2-infected AD cases compared to neurological controls (*Figure 1C*) revealed 38,021 genes and showed 813 genes were upregulated and 611 genes were downregulated using the same cutoff for significance (*Figure 1D*).

Transcriptional profiles of AD and SARS-COV-2 cases from the HF were assessed (p<0.01 and log2 FC > 1) to determine gene expression compared to neurological control cases (*Figure 1E–G*) using 39,901 genes for AD and SARS-CoV-2 cases (*Figure 1E and F*), and 35,527 genes for the SARS-CoV-2-infected AD cases (*Figure 1G*). In the AD group, 304 genes were upregulated and 214 genes were downregulated (*Figure 1H*). In the SARS-CoV-2 group, 230 genes were upregulated and 464 genes were downregulated (*Figure 1H*). Additionally, in the SARS-CoV-2-infected AD group, 466 genes were upregulated and 582 genes were downregulated (*Figure 1H*).

## Similarity of gene expression in cortical BA9 and HF regions

In order to determine the relative similarity of gene expression changes induced by disease states, a collection of genes known to be differentially expressed due to either SARS-CoV-2 infection or AD were examined across three individual differential comparisons (*Figure 2A and B*). While all three comparisons (AD versus control, SARS-CoV-2 versus control, and SARS-CoV-2-infected AD versus control) showed generally the same trends, the AD and SARS-CoV-2-infected AD comparisons were strikingly similar in both BA9 (*Figure 2A*) and HF (*Figure 2B*).

Rank-rank hypergeometric overlap (RRHO) analysis comparing SARS-CoV-2 and AD cases also reveals a positive correlation between the AD and SARS-CoV-2 groups in the cortical BA9 (*Figure 2—figure supplement 1*) as well as in the HF (*Figure 2—figure supplement 2*), although to a lesser extent than in the cortical BA9. We then filtered genes using Ingenuity Pathway Analysis software (IPA) to determine the number of shared differentially regulated genes among the AD, SARS-CoV-2, and SARS-CoV-2-infected AD groups within each tissue region (p<0.05 and absolute z > 0.0001). Comparing these shared genes within the cortical BA9, we find that AD, SARS-CoV-2, and SARS-CoV-2-infected AD individuals shared 410 upregulated genes and 269 downregulated genes from 679 expressed genes (*Figure 2C* and *Source data 1*). In the HF, we found that the three groups share 211 upregulated genes and 148 downregulated genes from 359 differentially expressed genes (*Figure 2D* and *Source data 2*).

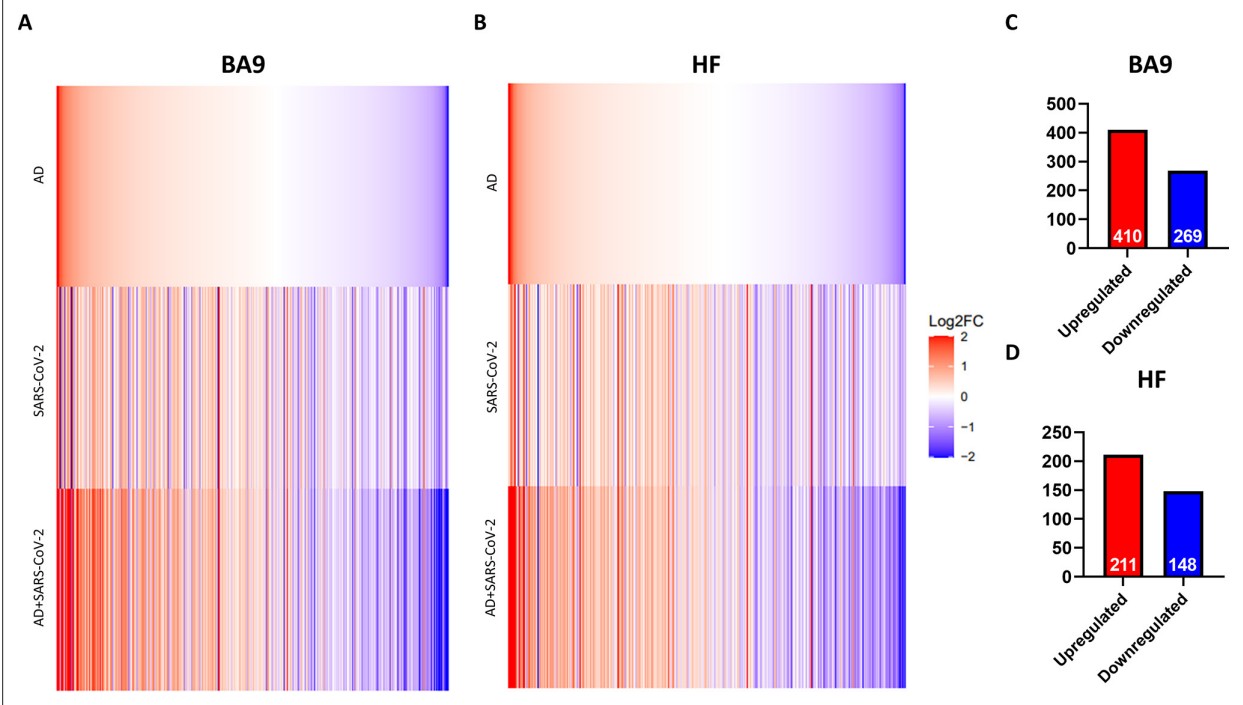

**Figure 2.** Similarity of gene expression in cortical Brodmann area 9 (BA9) and hippocampal formation (HF) in SARS-CoV-2 and SARS-CoV-2-infected Alzheimer's disease (AD) relative to AD individuals based on log2 fold-change (log 2FC). (**A, B**) Genes known to be differentially expressed due to either SARS-CoV-2 infection or AD were examined with respect to their gene expression changes, with the given gene set being selected. The log2 FC of the genes for three individual differential comparisons (AD versus control, SARS-CoV-2 versus control, and SARS-CoV-2-infected AD versus control) are shown for cortical BA9 (**A**) and HF (**B**), respectively. (**C, D**) Using Ingenuity Pathway Analysis (IPA) as a filtering system for genes within the database, we compared the similarity in expression of genes present in all datasets (SARS-CoV-2, AD and SARS-CoV-2-infected AD cases) in both the cortical BA9 (**C**) and HF (**D**).

The online version of this article includes the following figure supplement(s) for figure 2:

**Figure supplement 1.** Rank-rank hypergeometric overlap (RRHO) analysis of SARS-CoV-2/control and Alzheimer's disease (AD)/control, each containing (39,902 differentially expressed genes [DEGs]) revealed that gene regulation between the AD/control and SARS-CoV-2/control groups showed a positive correlation in the cortical Brodmann area 9 (BA9) region.

**Figure supplement 2.** Rank-rank hypergeometric overlap (RRHO) analysis of SARS-CoV-2/control and Alzheimer's disease (AD)/control, each containing (39,902 differentially expressed genes [DEGs]) revealed that gene regulation between the AD/control and SARS-CoV-2/control groups showed a positive correlation in the HP region.

## Pathway activation of SARS-CoV-2 and AD groups

IPA was employed to further characterize canonical pathways in AD, SARS-CoV-2, and SARS-CoV-2-infected AD groups compared to the neurological controls. Within the cortical BA9 region (*Figure 3A*), for example, several pathways, including neuroinflammation, TREM1, and cell senescence, show increased activation in the AD, SARS-CoV-2, and SARS-CoV-2-infected AD groups. TREM1 signaling, the most highly activated pathway in this dataset (*Figure 3A and B*), is expressed primarily on myeloid cells, such as macrophages and microglia, and is involved in pro-inflammatory immune responses (*de Oliveira Matos et al., 2020*). Although age-associated cellular senescence (*Figure 3A*) is a natural process where telomeres shorten over time, this process also occurs during cellular stress due to inflammation, including that caused by viral infections, leading to a senescence-associated secretory phenotype such as metalloproteinases (MMPs), and inflammatory cytokines (*Lee et al., 2021*; *Coppé et al., 2010*). Another pathway of interest, SNARE, shows decreased activation in the AD and SARS-CoV-2 groups (*Figure 3A*), with the most significant reduction seen in the SARS-CoV-2-infected AD cases group. SNARE proteins play an essential role in neurotransmitter release, and altered function is implicated in the pathophysiology of neurodegenerative diseases such as AD, where SNARE proteins affect β-amyloid (Aβ) accumulation and cytoplasmic transport of neurofibrillary tangles (*Margiotta, 2021*).

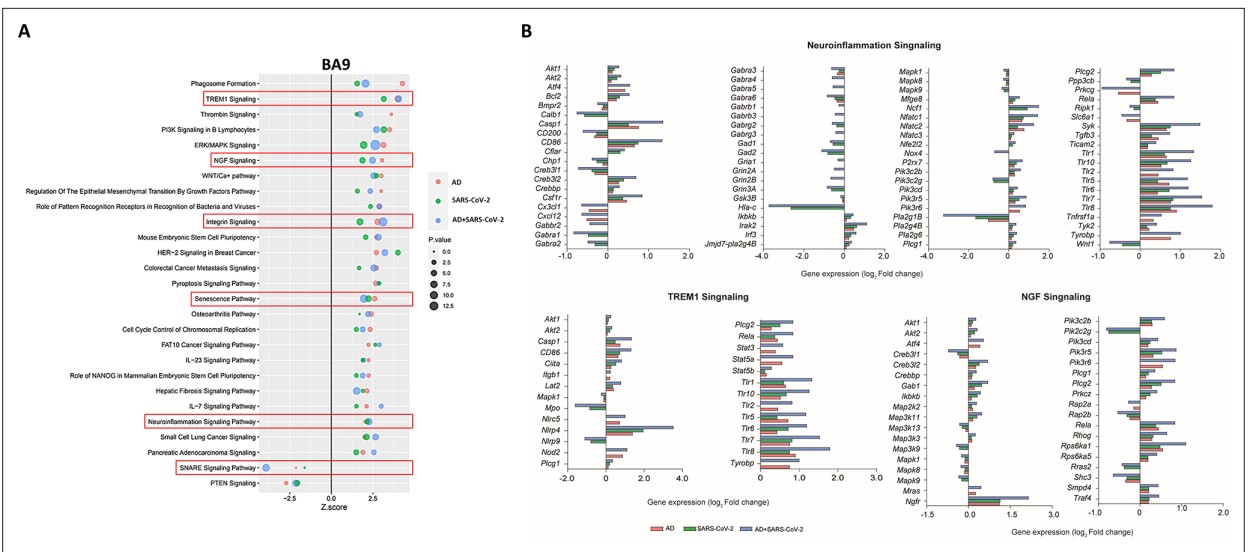

**Figure 3.** Changes in signaling pathways within the cortical Brodmann area 9 (BA9). (**A**) The similarities of canonical pathways reveal the top regulated canonical pathways within the cortical BA9 in Alzheimer's disease (AD), SARS-CoV-2, and SARS-CoV-2-infected AD cases in reference to the control. Activation score (Z-score) is shown on the X-axis, while the pathways are indicated on the Y-axis. The color of the points indicates the Ingenuity Pathway Analysis (IPA) comparison, while the size of the point represents the -log10 p-value of the IPA comparison, with the larger points indicating the lowest p-values. (**B**) The predicted gene regulation of the Neuroinflammation, TREM1, and nerve growth factor (NGF) pathways indicates that AD (red bar), SARS-CoV-2 (green bar), and SARS-CoV-2-infected AD groups (blue bar) have similar expression in key inflammatory and neuronal pathways, with log2 fold-change (log2 FC) shown on the X-axis.

Within the HF (*Figure 4A*), for example, interleukin-8 (IL-8) another neuroinflammatory pathway, and ciliary neurotrophic factor (CNTF) signaling is upregulated in AD, SARS-CoV-2, and SARS-CoV-2-infected AD groups, while cAMP response element-binding protein (CREB) is also upregulated in the SARS-CoV-2 groups but downregulated in AD-only (*Figure 4A*). IL-8 has several important roles, including endothelial cell migration and chemoattraction of neutrophils (*Manda-Handzlik and*

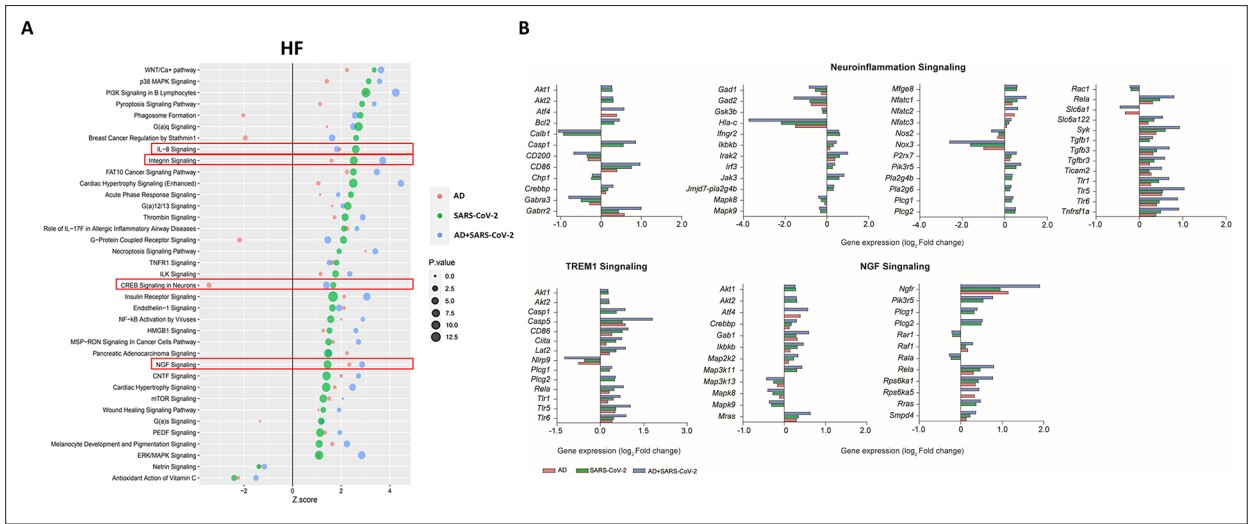

**Figure 4.** Changes in signaling pathways within the hippocampal formation (HF). (**A**) The similarities of canonical pathways reveal the top regulated canonical pathways were compared within the HF in Alzheimer's disease (AD), SARS-CoV-2, and SARS-CoV-2-infected AD cases in reference to the control. The X-axis represents the activation score (Z-score), while the Y-axis indicates the pathways. The color of the points reflects the Ingenuity Pathway Analysis (IPA) comparison, while the size of each point corresponds to the -log10 p-value of the IPA comparison, with larger points indicating lower p-values. (**B**) The predicted gene regulation of the Neuroinflammation, TREM1, and nerve growth factor (NGF) pathways indicates that AD (red bar), SARS-CoV-2 (green bar), and SARS-CoV-2-infected AD groups (blue bar) have similar expression in key inflammatory and neuronal pathways. The X-axis represents log2 fold-change (log2 FC).

*Demkow, 2019*). One way IL-8 aids cell migration is by enhancing the expression of molecules such as MMP-2, MMP-9, involved in BBB integrity and induction of neuronal apoptosis, and VEGF-A, involved in vascular permeability and angiogenesis, thus having a potential effect on vascular damage (*Lugano et al., 2020*; *Jian et al., 2018*; *Licht and Keshet, 2013*). CNTF signaling aids in the prevention of neuronal degeneration after injury and is neuroprotective in diseases such as multiple sclerosis (MS) and amyotrophic lateral sclerosis (ALS) (*Pasquin et al., 2015*). Interestingly, CREB signaling modulates processes in consolidating memory and information processing and is inhibited in AD (*Amidfar et al., 2020*). Consistent with these findings, we also see a reduction in CREB signaling in the AD individuals; however, this effect is reversed in SARS-CoV-2-infected AD individuals in the HF. It is important to note that TREM1 signaling is upregulated in two of the three sample groups (SARS-CoV-2 and SARS-CoV-2-infected AD cases), but no predictions for this pathway occur in the AD group. As such, this pathway is not included in the analysis within the HF (*Figure 4A and B*).

Additionally, upregulation of integrin signaling and nerve growth factor (NGF) signaling occur in both cortical BA9 and HF tissue (*Figures 3B and 4B*) in all SARS-CoV-2, AD and SARS-CoV-2 infected AD groups. Neuronal function shown by the upregulation of NGF signaling is critical for the survival of neurons, and alterations of NGF signaling has been implicated in neurodegenerative disorders such as AD (*Eu et al., 2021*). Integrin signaling is used for a diverse array of functions within the CNS via cell-to-cell and cell-to-extracellular matrix interactions (*Ikeshima-Kataoka et al., 2022*).

## Regulation of inflammatory microglia responses in cortical BA9 and HF

Region-matched formalin-fixed paraffin-embedded (FFPE) tissue from the contralateral hemisphere of cortical BA9 and HF used for transcriptional studies underwent immunohistochemical (IHC) analysis for evidence of microglial activation (Iba-1), the presence of SARS-CoV-2 (SARS-CoV-2 nucleocapsid), and vascular integrity, assessed via hypoxia-inducible factor-1α subunit (HIF-1α).

Neuroinflammation was assessed with a pan-microglia marker, Iba-1, which is upregulated by microglia in the context of inflammation and reveals morphological alterations associated with their activation state (*Figure 5*). Compared to neurological controls, microglia in AD, SARS-CoV-2, and SARS-CoV-2-infected AD cases are more numerous and display an activated phenotype, with retracted, thickened processes and enlarged somas (*Figure 5A–D*). Microglia also appear to accumulate around blood vessels in the context of AD and SARS-CoV-2 groups but not in unaffected neurological controls brain (*Figure 5E–H*), suggesting that factors at the level of the BBB may participate in microglial activation. Microglial nodules, which are commonly observed in neuroinflammatory disease, were seen in all conditions but appeared larger and more frequently in infection compared to AD-only and controls (*Figure 5I–L*). When detectable, SARS-CoV-2 appears to be restricted to the endothelium (*Figure 5G, H, K and L*) and did not colocalize with Iba-1 or glial fibrillary acidic protein (GFAP; data not shown), suggesting that neither microglia nor astrocytes harbor productive virus. Nonbiased quantitation revealed an overall increase in the number of microglia in the HF and cortical BA9 of patients with AD, SARS-CoV-2, and SARS-CoV-2-infected AD cases. A statistically significant increase in the number of microglia occurs in AD alone in the HF and SARS-CoV-2-infected AD cases in both the HF and cortical BA9 (*Figure 5M and O*). An increase in the frequency of nodular lesions was seen with SARS-CoV-2 infection, with and without AD, in both brain regions assessed. However, a statistically significant increase in nodular lesions is present in SARS-CoV-2-infected AD cases (*Figure 5P–R*).

## Regulation of vascular damage using hypoxia-inducible factor-1α (HIF-1α) detection

Tissues were further assessed for possible hypoxia by IHC using an antibody against the α subunit of HIF-1, which is stabilized under hypoxic conditions. Upregulation and stabilization of HIF-1α are most pronounced in the context of SARS-CoV-2 infection, regardless of AD status in cortical BA9 (*Figure 6A–D*) and HF (*Figure 6E–H*). Nonbiased area positivity quantitation revealed that HIF-1α is only slightly elevated in cortical BA9 of AD patients compared to age-matched neurological controls (*Figure 6J*). In contrast, a greater increase occurs in the HF from SARS-CoV-2 individuals, with and without AD. SARS-CoV-2 only samples showed a significant increase in HIF-1α of cortical BA9 compared to controls (*Figure 6I–K*).

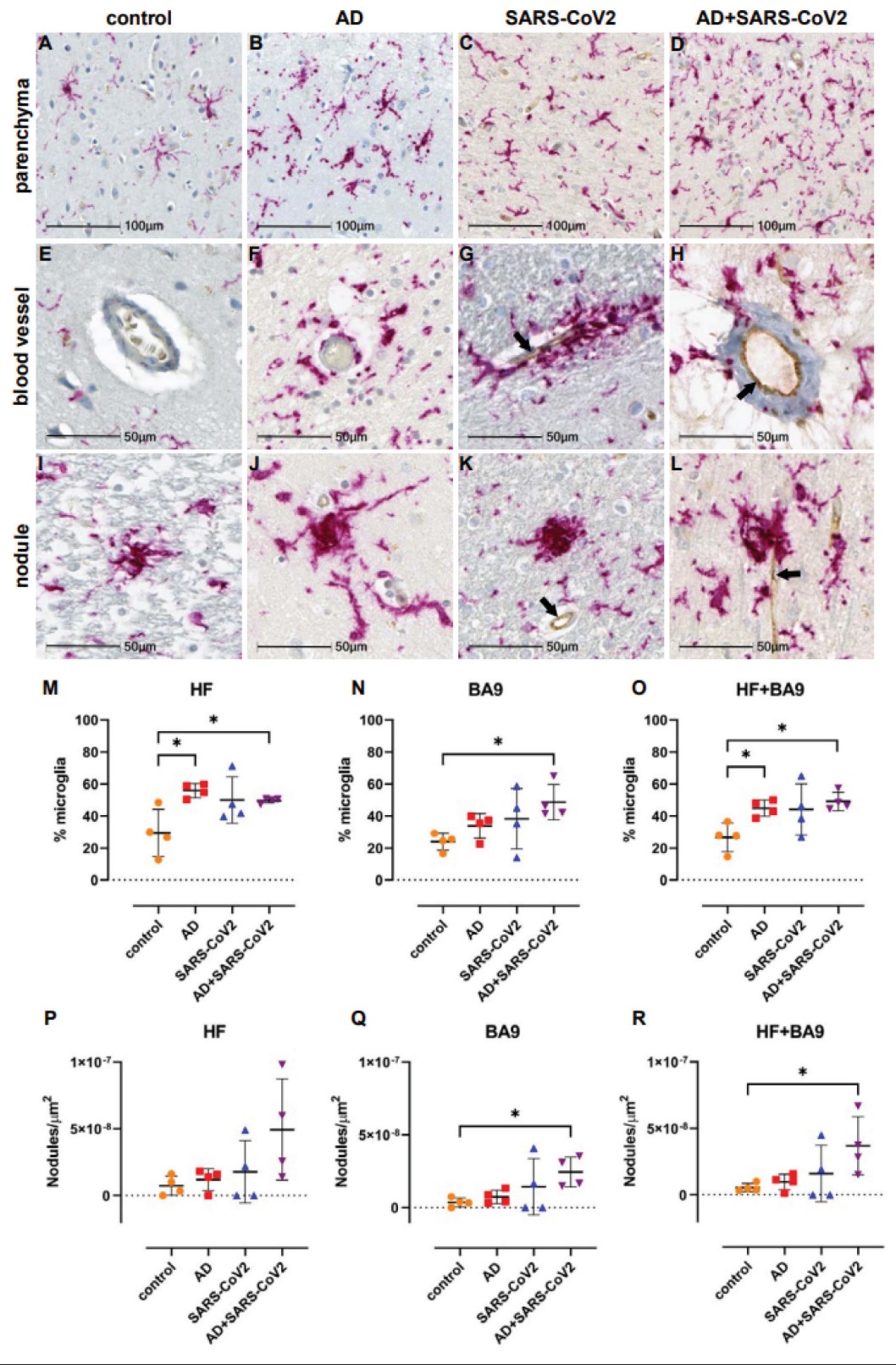

**Figure 5.** Microgliosis and nodular lesions in neurological controls, SARS-CoV-2, Alzheimer's disease (AD), and SARS-CoV-2-infected AD individuals. The level of microglial activation was assessed by immunohistochemical staining using anti-Iba-1 antibody with Vector Red. The presence of SARS-CoV-2 was assessed using an antibody against the virus nucleocapsid and visualized with DAB. (**B–D**) Parenchymal microglia are more frequent and

*Figure 5 continued on next page*

*Figure 5 continued*

highly activated in the context of infection with and without AD, as shown by thickened processes, enlarged soma, and loss of individual cellular domain, compared to age-matched controls (**A, E**). When present, SARS-CoV-2 localizes to the blood vessel endothelium (black arrows; **G, H, K, L**). (**F–H**) Microglia appear to gather around blood vessels in disease but do not form cuffs. (**I–L**) Nodular lesions are seen in most cases assessed, regardless of disease status; however, they appear larger and more frequent in the context of disease (**J–L**), compared to controls (**I**). (**M–O**) A multiplex algorithm was used to count all cells, using DAPI+ nuclei, with HALO and calculate percent frequency of Iba-1+ microglia. Each point on the graphs shown represents the average finding per total brain area for each subject. (**M**) There was upregulation of Iba-1 in the hippocampus compared to the cortical Brodmann area 9 (BA9) region for most cases, though there is a significantly higher ratio of microglia relative to other cells when comparing the AD and SARS-CoV-2-infected AD cases to the control group shown. (**N**) This trend seems to hold true for cortical BA9 region as well, where the only groups with a significant difference in microglia ratios are the control and SARS-CoV-2-infected AD cases shown. (**O**) When all cases have both regions averaged, the level of microgliosis is shown to be higher in the AD and SARS-CoV-2-infected AD cases compared to control. (**P–R**) Graphs show the normalized counts of microglial nodule frequency. (**P**) A higher frequency of nodules is seen in the HF; however, the difference between groups did not reach statistical significance. (**Q**) In contrast, fewer nodules are seen in cortical BA9, overall. A statistically significant higher number of lesions are seen in SARS-CoV-2-infected AD cases compared to control cortical BA9, suggesting greater inflammation in the cortical BA9 of patients with both AD and SARS-CoV-2 infection. (**R**) Significance was maintained between the control and SARS-CoV-2-infected AD groups when the average was taken for both brain regions per case. Statistics were performed with a two-tailed Mann–Whitney *U*-test. *p<0.05. Data are expressed as mean ± SEM.

## Discussion

The finding from our study suggests that SARS-CoV-2 and AD-infected individuals share similar alterations of regulatory patterns of immune-inflammatory pathways and pathways involved with cognition as suggested by recent meta-analyses showing shared neuroinflammation and microvascular injury, in particular AD and SARS-CoV-2-infected individuals (*Zhou et al., 2021*). Additionally, Zhou and colleagues found a significant overlap in cerebrospinal fluid (CSF) monocytes and markers in AD and COVID-19, which also occurs in our dataset (*Zhou et al., 2021*). The similarities of transcriptional profiles and cellular pathophysiology in SARS-CoV-2 and SARS-CoV-2-infected individuals support the potential role of SARS-CoV-2 infection on the CNS, leading to neuro-PASC symptoms such as brain fog and memory loss.

Notably, our study identifies similar neuroinflammatory profiles in SARS-CoV-2, AD, and AD SARS-CoV-2-infected individuals at the transcriptional and cellular levels in both the cortical BA9 and HF brain regions. This suggests that SARS-CoV-2 generates a similar neuroinflammatory environment in neurodegenerative disorders like AD. This was highlighted by the regulation of TREM1, neuroinflammation, and cellular senescence/inflammatory pathways present in all groups and further established by the widespread microglial activation seen in AD and SARS-CoV-2-infected AD cases, and in particular, the nodular formation seen in SARS-CoV-2-infected AD cases. Microglia nodule formation is present in some neurodegenerative diseases, such as MS, and viral infections, such as herpes simplex virus (HSV) and human immunodeficiency virus (HIV) (*Tröscher et al., 2019*; *Fischer-Smith, 2001*). A potentially compounding finding is the nodular formation that is initially characterized by abundant presence of activated microglia and innate immune factors leading to Toll-like receptor (TLR) signaling and upregulations of inflammasome genes, leading to T cell stimulation and ultimately the destruction of neurons (*Tröscher et al., 2019*). This may be of particular concern in the SARS-CoV-2-infected AD cases, where we primarily observe increased nodular lesions. This suggests that SARS-CoV-2 further promotes neuroinflammation in AD, which likely advances the progression and severity of CNS disease in these individuals. Interestingly, a large retrospective study of patients 65 years or older revealed that patients with SARS-CoV-2 were at an increased risk for a new AD diagnosis within a year of their SARS-CoV-2 diagnosis, with the most significant risk seen among those 85 years and older (*Wang et al., 2022*). This may underscore the critical role of preexisting inflammation in the brain, which is seen in the context of 'normal' aging, in promoting or advancing AD progression.

Our findings also support the notion that SARS-CoV-2 may cause cognitive deficits via regulation of pathways associated with cognition and neuroinflammation. Here, we show changes in the transcriptional regulation of SNARE and NGF pathways, suggesting impaired neuronal health and function, presumably negatively impacting cognitive function. We also observed potential damage to the

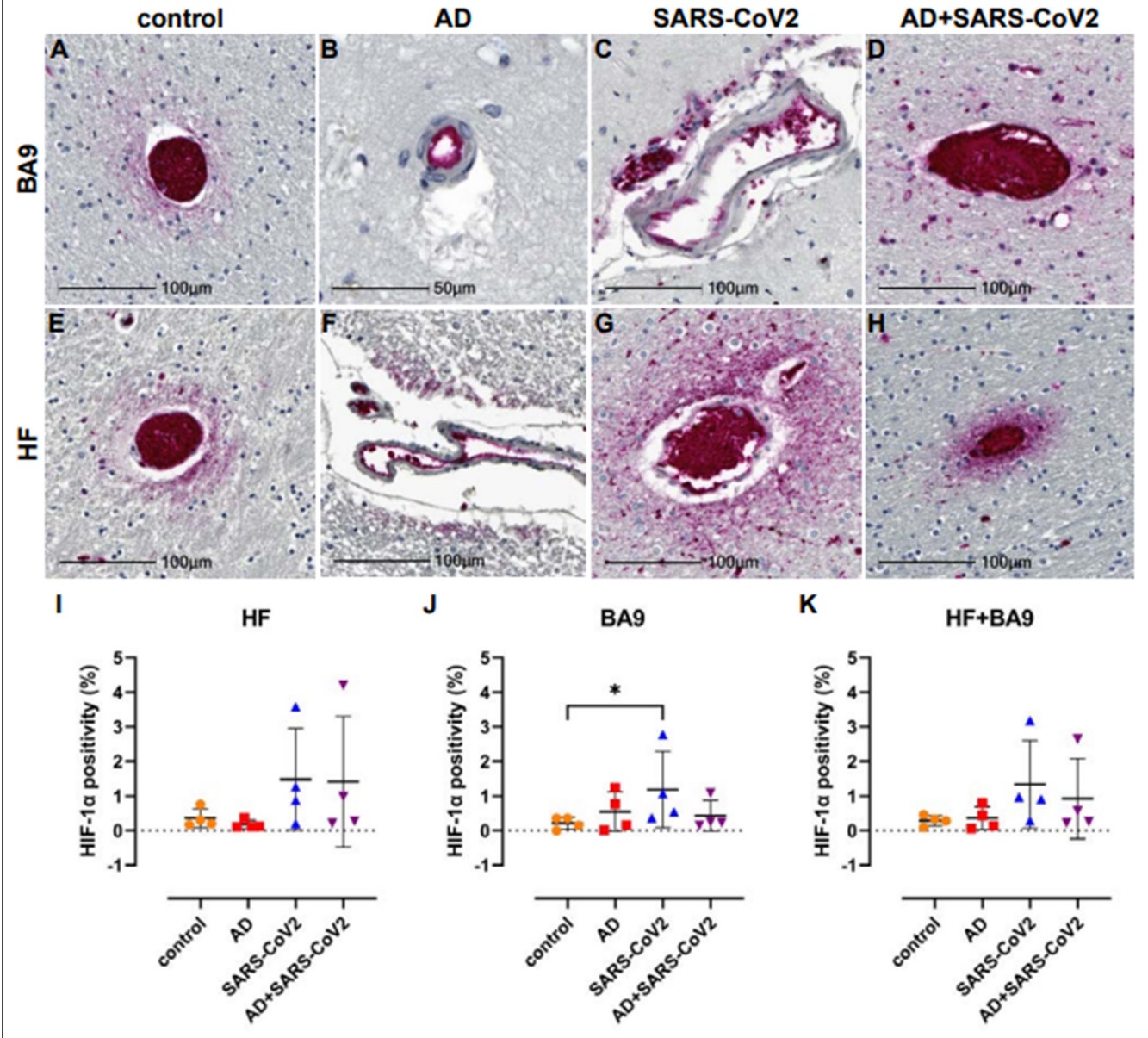

**Figure 6.** Hypoxia-inducible factor-1 alpha subunit (HIF-1α) in neurological controls, SARS-CoV-2, Alzheimer's disease (AD), and SARS-CoV-2-infected individuals. (**A–H**) Representative images demonstrate HIF-1α immunopositivity around the vasculature that extends into the parenchyma in brain of patients with SARS-CoV-2 infection (**C, D, G, H**), regardless of AD status. Comparatively, this is seen less frequently and does not extend significantly into the parenchyma in brain of age-matched controls (**A, E**). In AD only, positivity was most often observed in epithelium with no or minimal extension into the brain parenchyma (**B, F**). (**I–K**) Graphs show the total average percentage of tissue positive for HIF-1α for each individual subject, as defined using a HALO area algorithm for detection of Vector Red intensity over the whole section. Although statistical significance between groups was not reached in the hippocampal formation (HF) (**I**) or averaged group comparisons (**K**), increased HIF-1α expression is seen in the SARS-CoV-2-infected patients, with and without AD, compared to age-matched controls. An increase in area positivity is seen in all groups in the cortical Brodmann area 9 (BA9) region (**J**), compared to nonaffected controls; however, statistical significance is only seen with the SARS-CoV-2 group. Statistics were performed with a two-tailed Mann–Whitney $U$-test. *p<0.05. Data are expressed as mean ± SEM.

vasculature via increased regulation of HIF-1α, integrin signaling, and IL-8 signaling. It is important to note the possibility of vascular damage due to ventilation prior to death in SARS-CoV-2-infected individuals; however, vascular damage by SARS-CoV-2, as assessed by HIF-1α, is observed in non-human primates that were euthanized at designated end-of-life time points precluding breathing intervention (*Rutkai et al., 2022*). These findings may point to a possible route for lymphocyte transmigration following chemoattractant gradients such as IL-8 and enhancement of MMPs and VEGF, which are

aided by integrins such as cellular adhesion molecules such as ICAM-1 and are implicated with inflammatory signal transduction (*Patarroyo et al., 1990*; *Bui et al., 2020*). This process leads to T cell activation, which may be responsible for the observed inflammatory environment. Our transcriptional data showed an abundant upregulation of TLRs (TLRs 1, 5, 7, and 8) related to the inflammasome. Further analysis of this dataset also indicated the potential role for T cells through IL-7 pathway activation and CD86 upregulation; however, we did not confirm changes in the number of T cells, and no evidence of leukocyte infiltration into the CNS compartment was observed in any disease state in this investigation. Future studies will be required to determine the role of leukocytes in the observed pathophysiology.

Although neuroinflammation and vascular injury were prominent features of SARS-CoV-2 and SARS-CoV-2-infected AD cases brain pathology, the direct role of the virus is unclear. We did detect SARS-CoV-2 nucleocapsid in some SARS-CoV-2 brain tissues that appear to be restricted to the vasculature. This finding is supported by other studies suggesting that SARS-CoV-2 is sporadically present in brain tissue (*Serrano et al., 2021*). Importantly, we only investigated CNS regions with the greatest significance in AD. These regions may be less prone to SARS-CoV-2 infection than others, such as the olfactory bulb and tract, where olfactory neurons are proposed to be infected through viral spread from olfactory epithelium (*Burks et al., 2021*). Our findings of the viral nucleocapsid limited to the endothelium suggest the hematological spread of SARS-CoV-2 to the CNS that may not extend to the neurons in the cortical BA9 and HF yet is still capable of inducing widespread inflammation in the brain. Even in the absence of a detectable virus in the neurons or neural cells, SARS-CoV-2 may impact cognitive dysfunction through TREM1 activation of the NLRP3 inflammasome and subsequent pyroptosis, where pro-caspase 1 cleavage and subsequent cleavage and activation of IL-1β, IL-18, and gasmerdin D pore formation in cells ultimately lead to pyroptosis (*Parodi-Rullán et al., 2021*; *Xu et al., 2021*). NLRP3 activation is reported in AD and COVID-19, and is suggested by the formation of microglial nodules demonstrated in these cases (*Lünemann et al., 2021*; *Herman and Pasinetti, 2018*; *Campbell et al., 2021*).

SARS-CoV-2 is not the first virus to be implicated in cognitive dysfunction. This is a facet shown with other viruses such as HIV, HSV, and Epstein–Barr virus (EBV) (*Smail and Brew, 2018*; *Sun et al., 2022*). While this study cannot predict the outcome of disease progression in COVID-19 survivors, the present findings that SARS-CoV-2 infection can recapitulate AD-type transcriptional and cellular neuroinflammatory patterns among other in a very short time frame make it critical to understand how SARS-CoV-2 impacts long-term cognition. The increase in nodular formation present in SARS-CoV-2-infected AD cases tissue also demonstrates a critical need to functionally determine potentially synergistic effects of AD and SARS-CoV-2. This is underscored by the prevalence of cognitive dysfunction seen among neuro-PASC patients, making it imperative that the link between SARS-CoV-2 and cognition be intensively investigated to identify potential therapeutic strategies for halting cognitive decline in these individuals. Collectively, our results demonstrate several key areas of overlap between the neurological effects of SARS-CoV-2 infection and AD. These findings may help inform decision-making regarding therapeutic treatments in patients with COVID-19, especially those who may be at increased risk of developing AD.

## Materials and methods

### Patients

Brain tissue was collected from the cortical BA9 and HF of four SARS-CoV-2 cases, four AD cases, and four non-SARS-CoV-2 or AD autopsies. With each case, one hemisphere of the brain was FFPE and the other hemisphere was frozen to generate fresh-frozen blocks. FFPE tissue was used for microscopy, and region-matched fresh-frozen tissue was used for sequencing. Tissue was collected in accordance with IRB-approved guidelines and regulations by the Brain Bank at Mount Sinai, and clinical data, including cardiovascular and neurological conditions, were collected by the Department of Neurology.

### RNA sequencing

Samples collected and homogenized in RNAzol RT (Molecular Research Center, Inc) were then processed using the Zymo Clean and concentrator Kit (Zymo Research) to collect total RNA following the manufacturer's protocols. cDNA library construction and sequencing were conducted by Genewiz

(Azenta Life Sciences). Total RNA samples were quantified using Qubit 2.0 Fluorometer (Life Technologies, Carlsbad, CA), and RNA integrity was checked with 4200 TapeStation (Agilent Technologies, Palo Alto, CA). Samples were treated with TURBO DNase (Thermo Fisher Scientific, Waltham, MA) to remove DNA contaminants, followed by rRNA depletion using QIAseq FastSelect−rRNA HMR kit (QIAGEN, Germantown, MD), conducted following the manufacturer's protocol. RNA sequencing libraries were constructed with the NEBNext Ultra II RNA Library Preparation Kit for Illumina by following the manufacturer's recommendations, where enriched RNAs were fragmented for 15 min at 94°C. First-strand and second-strand cDNA were subsequently synthesized. cDNA fragments were end repaired and adenylated at 3' ends, and universal adapters were ligated to cDNA fragments, followed by index addition and library enrichment with limited cycle PCR. Sequencing libraries were validated using the Agilent Tapestation 4200 (Agilent Technologies) and quantified using Qubit 2.0 Fluorometer (Thermo Fisher Scientific), as well as by quantitative PCR (KAPA Biosystems, Wilmington, MA). The sequencing libraries were multiplexed and clustered on one lane of a flowcell. After clustering, the flowcell was loaded on the Illumina HiSeq 4000 instrument according to the manufacturer's instructions. The samples were sequenced using a 2 × 150 Pair-End (PE) configuration.

Image analysis and base calling were conducted using the HiSeq Control Software (HCS). Raw sequence data (.bcl files) generated from Illumina HiSeq was converted into FASTQ files and de-multiplexed using Illumina's bcl2fastq 2.17 software. One mismatch was allowed for index sequence identification. After investigating the quality of the raw data, sequence reads were trimmed to remove possible adapter sequences and nucleotides with poor quality using Trimmomatic v.0.36. The trimmed reads were mapped to the GRCh38 reference genome available on ENSEMBL using the STAR aligner v.2.5.2b. BAM files were generated as a result of this step. Unique gene hit counts were calculated by using feature Counts from the Subread package v.1.5.2. Only unique reads that fell within the exon regions were counted.

## Immunohistochemistry

Tissue was cut at 6 µm on a Leica Microtome for immunohistochemistry that was performed on FFPE brain sections as described previously (*Tavazzi et al., 2014*). Iba-1 staining was conducted on a Ventana Benchmark using OptiView and UltraView detection kits provided by Roche (Roche Molecular Systems, Inc). Sections were deparaffinized in xylene and rehydrated through an ethanol series ending in distilled water. Heat-mediated antigen retrieval was carried out in a vacuum oven with Tris-EDTA buffer (10 mM Trizma base, 1 mM EDTA, 0.05% Tween 20, pH 9.0) or sodium citrate buffer (10 mM sodium citrate, 0.05% Tween 20, pH 6.0). All washes were performed using Tris buffered saline containing Tween 20 (TTBS; 0.1 M Trizma base, 0.15 M NaCl, 0.1% Tween 20, pH 7.4). Following antigen retrieval, tissues were blocked with 20% normal horse serum. Titrated primary antibodies included anti-HIF-1α (mouse mgc3, 1:400, Abcam, ab16066). Tissues were incubated with primary antibody overnight at room temperature and detected using the appropriate biotinylated secondary antibody (1:200, Vector Labs, BA-1100, BA-2000) and alkaline phosphatase-Vector Red according to the manufacturer's instructions (Vector Labs). Tissues were counterstained with Mayer's hematoxylin and coverslipped.

## Imaging and quantitation

Slides were scanned with the Axio Scan.Z1 digital slide scanner (Zeiss). Brightfield images were acquired using HALO (Indica Labs, v3.4.2986.151). Figures were created in Photoshop (Adobe, v23.5.1) by brightness and contrast adjustments applied to the entire image.

Threshold and multiplex analyses were performed with HALO algorithms for nonbiased quantitation of proteins of interest without processing, as described previously (*Burks et al., 2021*). For microglia quantitation, hematoxylin-stained nuclei were used to quantify the total number of cells and those with Vector Red intensity above a rigorous threshold (Iba-1+). Microglia frequency is reported as the percentage of total nuclei in the tissue section assessed for each individual subject. To assess frequency of nodular lesions, all Iba-1-stained tissues were viewed in HALO and nodular lesions made up of three or more Iba-1+ microglia in contact with one another were counted as a single nodule. The total number of nodules for each tissue is reported per tissue area for each individual patient. Quantitation of HIF-1α was performed using an area quantification algorithm for Vector Red intensity. Annotations were drawn to outline the tissue and exclude empty spaces and glass. The annotated

area was analyzed for overall quantity of Vector Red positivity per micron$^2$ of tissue and reported for each individual patient. Two-tailed Mann–Whitney $U$-tests were performed with GraphPad Prism software, v9.3.1. Data are expressed as mean ± SEM. p-Values≤0.05 are considered significant.

## Bioinformatics data access and analysis

Sample similarity was assessed with PCA on VST-transformed expression values. Genes were filtered to remove lowly expressed genes, defined as fewer than five samples showing a minimum read count of 1 read, prior to performing differential analysis with DESeq2, in R (4.2.0; *Varet et al., 2016*). The transcriptional profiles of SARS-CoV-2 and AD postmortem cases from the cortical BA9 and HF were assessed by pairwise comparison to the neurologically healthy control cases. Multiple testing correction was performed for all comparisons; however, due to the relatively small number of genes surpassing a threshold of 0.05, a nominal p-value was used in the selection of differential genes for all downstream analyses. Genes were assigned as differentially expressed if the nominal p-value was <0.05 and the absolute log2FC exceeded 1. An additional round of differential expression testing was performed using a model containing covariates (COVID status, Alzheimer's status, gender, and brain region) to extract the transcriptional effects of the individual covariates. Biological pathways and key regulators impacted by disease were identified using QIAGEN IPA (QIAGEN Inc, version 73620684; *Krämer et al., 2014*). Genes with a threshold of p<0.05 were used as input for IPA. The relative similarities of transcriptional changes in the DESeq2 comparisons were assessed using Rank-rank hypergeometric overlap (RRHO2) analysis.

## Acknowledgements

The authors thank the Neuropathology Brain Bank and Research CoRE of the Icahn School of Medicine at Mount Sinai for postmortem samples. Giulio M Pasinetti holds a Senior VA Career Scientist Award. The views expressed in this article are those of the authors and do not necessarily reflect the position or policy of the Department of Veterans Affairs or the United States government. The research was supported by Unrestricted Funds and Altschul Foundation (GMP) National Institutes of Health Office of Research Infrastructure Programs grant P51OD01110 (TF) The content is solely the responsibility of the authors and does not necessarily represent the official views of the National Institutes of Health and Veteran Administration

## Additional information

### Competing interests

Nathalie Jette: receives grant funding paid to her institution from NINDS (NIH U24NS107201, NIH IU54NS100064, 3R01CA202911-05S1, R21NS122389, R01HL161847). Some of these grants are COVID-19 related but focus on the neuroimaging findings. The other authors declare that no competing interests exist.

### Funding

No external funding was received for this work.

### Author contributions

Elizabeth Griggs, Conceptualization, Data curation, Formal analysis, Validation, Investigation, Visualization, Methodology, Writing – original draft, Writing – review and editing; Kyle Trageser, Brian Mathew, Grace Van Hyfte, Investigation; Sean Naughton, Investigation, Writing – review and editing; Eun-Jeong Yang, Validation; Linh Hellmers, Li Shen, Visualization; Nathalie Jette, Supervision, Investigation, Writing – review and editing; Molly Estill, Investigation, Visualization, Writing – original draft; Tracy Fischer, Supervision, Investigation, Visualization, Writing – original draft, Writing – review and editing; Giulio Maria Pasinetti, Conceptualization, Funding acquisition

### Author ORCIDs

Eun-Jeong Yang (iD) http://orcid.org/0000-0003-3190-8604
Giulio Maria Pasinetti (iD) http://orcid.org/0000-0002-1524-5196

### Ethics

Human subjects: While no living humans or animals were used for these studies, we performed studies using human postmortem tissue in accordance with IRB-approved guidelines and regulations at Mount Sinai.

### Decision letter and Author response

Decision letter https://doi.org/10.7554/eLife.86333.sa1
Author response https://doi.org/10.7554/eLife.86333.sa2

## Additional files

### Supplementary files

• Supplementary file 1. Table 1. Demographics of postmortem cases. Numbers presented are the average per group and standard deviation PMI is postmortem interval (n = 4); OtD is onset to death. ABC is a multipoint measurement where A is a measure of amyloid β deposition, B is a measure of neurofibrillary degeneration based on the Braak score, and C is scored based on neuritic plaques outlined by the Consortium to Establish a Registry for Alzheimer's Disease diagnosis (CERAD).

• Supplementary file 2. Table 2. Blood chemistry and symptoms. Symptoms are presented as a percentage of the group (n = 4); chemistry numbers are presented as the average per group and standard deviation, based on maximum values collected for single values and the minimum and maximum range collected in cells with two values. Bolded values are implicated as indicators of severe SARS-CoV-2 outcomes; ALP, alkaline phosphatase; ALT, alanine aminotransferase; AST, aspartate transferase; CRP, c-reactive protein; ESR, erythrocyte sedimentation; INR, international normalized ratio; LDH, lactate dehydrogenase; WBC, white blood count; BUN, blood urea nitrogen.

• MDAR checklist

• Source data 1. Based on the DESeq, and filtered using IPA, genes that were expressed in all SARS-CoV-2 and AD in cortical BA9 datasets with an absolute expression of 0.0001 or greater and p<0.05 are presented.

• Source data 2. Based on the DESeq, and filtered using IPA, genes that were expressed in all SARS-CoV-2 and AD in HF datasets with an absolute expression of 0.0001 or greater and p<0.05 are presented.

### Data availability

Data generated for this study are accessible through NCBI's Gene Expression Omnibus (GEO) at GSE236562.

The following dataset was generated:

| Author(s) | Year | Dataset title | Dataset URL | Database and Identifier |
|---|---|---|---|---|
| Griggs E, Trageser K, Naughton S, Yang E-J, Mathew B, Van Hyfte G, Hellmers L, Jette N, Estill M, Shen L, Fischer T, Pasinetti GM | 2023 | Recapitulation of pathophysiological features of AD in SARS-CoV-2 -infected subjects | https://www.ncbi.nlm.nih.gov/geo/query/acc.cgi?acc=GSE236562 | NCBI Gene Expression Omnibus, GSE236562 |

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
