## [Editor Report]

This study compares neuropathological changes in postmortem brain samples from patients with SARS-CoV-2 infection, Alzheimer's disease, both SARS-CoV-2 and Alzheimer's disease, and demographically matched controls. This is an important comparison that expands information on SARS-CoV-2 effects in the brain. Solid results are presented to support the main claims.

---

## [Decision Letter]

**Decision letter after peer review:**

Thank you for submitting your article "Recapitulation of pathophysiological features of AD in SARS-CoV-2 infected subjects" for consideration by *eLife*. Your article has been reviewed by 2 peer reviewers, and the evaluation has been overseen by a Reviewing Editor and Laura Colgin as the Senior Editor. The reviewers have opted to remain anonymous.

Essential revisions:

*Reviewer #1 (Recommendations for the authors):*

The authors examined neuropathological changes in postmortem brain tissue samples from patients with SARS-CoV-2 infection, Alzheimer's disease, both SARS-CoV-2 and Alzheimer's disease, and demographically matched controls. This is an important comparison since there is still a lot that is unknown regarding SARS-CoV-2 effects in the central nervous system and there is concern that SARS-CoV-2 infection can accelerate cognitive decline or development of Alzheimer's disease in certain at-risk individuals.

The experimental tissue samples were obtained from a well-established Neuropathology Brain Bank at Mount Sinai Medical Center in New York. The authors examined brain tissue samples from the frontal cortex and the hippocampal formation, and analyzed transcriptional profiles using a volcano plot for distributions of gene transcripts of SARS-CoV-2 and AD cases compared to controls for thousands of genes. The authors identified genes with different expression according to a p-value of less than 0.01 and an absolute log fold change greater than 1, including upregulated and downregulated genes. The authors then utilized rank-rank hypergeometric overlap analysis and determined the number of shared differentially regulated genes among SARS-CoV-2, AD, and SARS-CoV-2/AD groups with a p value of less than 0.05 and an absolute z value of greater than 0.0001 for frontal lobe and hippocampal formation areas. Ingenuity Pathway Analysis software was then used to further characterize pathways with altered gene expression compared to controls, including pathways involved in inflammation, β-amyloid accumulation and neurofibrillary tangle transport, cell senescence, and integrin signaling and nerve growth factor signaling. Markers for microglial activation and hypoxia were also assessed.

The authors concluded that SARS-CoV-2 and AD cause similar alterations of regulatory patterns of immune-inflammatory pathways at the transcriptional and cellular levels. The authors also explained that the data suggests that SARS-CoV-2 infection may exacerbate neuroinflammation in Alzheimer's disease which could advance the progression and severity of the disease process in these individuals and may contribute to increased risk for development of Alzheimer's disease in susceptible individuals as well. The authors also make the point that their findings support the notion that SARS-CoV-2 may cause cognitive deficits through disruption of regulation of pathways associated with neuroinflammation and that they detected SARS-CoV-2 nucleocapsid in brain vasculature but not in neurons.

This is an excellent and timely study that highlights yet another potential contribution of viruses to the development of Alzheimer's disease pathophysiology. Additionally, the results of this study may be useful in improving understanding of how SARS-CoV-2 infection produces long-term cognitive dysfunction in certain individuals.

This is an interesting and excellent study. I think the information in this article would be useful to clinicians caring for patients with both SARS-CoV-2 and Alzheimer's disease. I am not really an expert on a lot of the methods that were used or this type of genetic analysis, but the conclusions seemed logical and well supported by the data.

*Reviewer #2 (Recommendations for the authors):*

The importance of SARS-CoV-2 mediated neuro-pathophysiological change in Alzheimer's disease is a very important subject. Elizabeth et al., investigated the common aspects of AD and COVID-19 based on the four conditions; SARS-CoV-2 infected, AD and AD individuals who became infected by SARS-CoV-2, compared to age- and gender-matched neurological cases by comparing transcriptional and cellular responses in the cortical Broadman area 9 (cortical BA9) of the frontal cortex and the hippocampal formation (HF); The authors reported increased Iba-1 positive microglial cell in conjunction with the morphological alterations in SARS-CoV-2 infected AD individuals. Similarly, they can observe HIF-1α is significantly upregulated in the context of SARS-CoV-2 infection in the same brain regions regardless of the AD status.

Although the study subject is very important, the conclusions are not sufficiently supported to advance our understanding of AD pathophysiology in the context of viral infection. Many of the observed effects could simply be blamed on the fact that neuroinflammation occurs with both AD and viral infections. No data address the effects of COVID-19 on AD-specific pathology – amyloid plaques and Tau-tangles. In addition, significant analytical details are missing, e.g. FDR values.

The study assesses the importance of COVID-19 and/or SARS-CoV-2 mediated neuro-pathophysiological change in Alzheimer's disease based on four conditions; SARS-CoV-2 infected, AD and AD individuals who became infected by SARS-CoV-2, compared to age- and gender-matched neurological cases by comparing transcriptional and cellular responses in the cortical Broadman area 9 (cortical BA9) of the frontal cortex and the hippocampal formation (HF); The authors reported that increased Iba-1 positive microglial cell in conjunction with morphological alterations in SARS-CoV-2 infected AD individuals. Similarly, they can observe HIF-1α is significantly upregulated in the context of SARS-CoV-2 infection in the same brain regions regardless of AD status. Although the study subject is important, the conclusions made are not yet sufficiently supported to advance our understanding of AD pathophysiology in the context of viral infection.

Since enhanced inflammatory activity is already expected in the context of viral infection, one of the prioritized and/or main questions answered should be centered on the two other AD two-hallmarks: amyloid-β plaque burden and neurofibrillary tangles. The big problem is that the emphasis is on pathology a priori expected with viral infection – neuroinflammation – which is also a major player in AD. Unfortunately, the study all but ignores the impact on the lesions that define AD – plaques and tangles. Thus, the central message of "recapitulated pathophysiological features of AD in SARS-CoV-2 infection could readily be attributed to viral infection, alone. In addition, significant analytical details are missing as listed below which makes it difficult to interpret the finding. So, while important, this study requires further analysis and clarification before being published.

Technical Concerns:

1. The authors selected the significant gene changes based on only p-value and fold change. However, it is essential that the authors include the "false discovery rate (FDR)" values, which would greatly affect the validity and reliability of the findings. So, it is highly recommended that authors perform weighted correlation network analysis (WGCNA) to dissect minimally which gene modules and/or traits are affected in SARS, AD and AD with SARS-CoV-2 infected cases (https://doi.org/10.1186/1471-2105-9-559).

2. Figure 1 Gene expression profiles are well described but would be great if the authors could dissect up and down pathways based on the DEG changes. Figure 1 itself is not informative enough to get insights, so it is highly recommended that the authors report what genes/pathways are affected based on their observations.

3. Figure 2 gene names are not described. Thus, it is hard to understand what is up or down based on the conditions.

4. Figure 3 It is interesting that TREM1 signaling significantly changed in AD and AD with SARS-CoV-2, but more important that the authors should compare AD with SARS-CoV-2 vs AD condition, as well.

5. Figure 3 and Figure 4 Need to address/discuss why are the transcriptomic data from two regions – BA9 and HF – are showing different results.

6. Figure 5 The detailed experimental procedures are needed. e.g. is one dot representing the one individual subject? Or one dot is from the one image?

7. Figure 5. Microglial cells are less activated in brain parenchyma while they are accumulating more significantly around vessels. Thus, compromised blood vessels may be affected by activated microglial cells and consequently lead to more immune cell infiltration. As such, it would be useful if the authors could address whether they observed any peripheral immune cell infiltration such as T cells, B cells, neutrophils, and/or monocytes.

---

## [Author Response]

Essential revisions:Reviewer #2 (Recommendations for the authors):The importance of SARS-CoV-2 mediated neuro-pathophysiological change in Alzheimer's disease is a very important subject. Elizabeth et al., investigated the common aspects of AD and COVID-19 based on the four conditions; SARS-CoV-2 infected, AD and AD individuals who became infected by SARS-CoV-2, compared to age- and gender-matched neurological cases by comparing transcriptional and cellular responses in the cortical Broadman area 9 (cortical BA9) of the frontal cortex and the hippocampal formation (HF); The authors reported increased Iba-1 positive microglial cell in conjunction with the morphological alterations in SARS-CoV-2 infected AD individuals. Similarly, they can observe HIF-1α is significantly upregulated in the context of SARS-CoV-2 infection in the same brain regions regardless of the AD status.Although the study subject is very important, the conclusions are not sufficiently supported to advance our understanding of AD pathophysiology in the context of viral infection. Many of the observed effects could simply be blamed on the fact that neuroinflammation occurs with both AD and viral infections. No data address the effects of COVID-19 on AD-specific pathology – amyloid plaques and Tau-tangles. In addition, significant analytical details are missing, e.g. FDR values.The study assesses the importance of COVID-19 and/or SARS-CoV-2 mediated neuro-pathophysiological change in Alzheimer's disease based on four conditions; SARS-CoV-2 infected, AD and AD individuals who became infected by SARS-CoV-2, compared to age- and gender-matched neurological cases by comparing transcriptional and cellular responses in the cortical Broadman area 9 (cortical BA9) of the frontal cortex and the hippocampal formation (HF); The authors reported that increased Iba-1 positive microglial cell in conjunction with morphological alterations in SARS-CoV-2 infected AD individuals. Similarly, they can observe HIF-1α is significantly upregulated in the context of SARS-CoV-2 infection in the same brain regions regardless of AD status. Although the study subject is important, the conclusions made are not yet sufficiently supported to advance our understanding of AD pathophysiology in the context of viral infection.Since enhanced inflammatory activity is already expected in the context of viral infection, one of the prioritized and/or main questions answered should be centered on the two other AD two-hallmarks: amyloid-β plaque burden and neurofibrillary tangles. The big problem is that the emphasis is on pathology a priori expected with viral infection – neuroinflammation – which is also a major player in AD. Unfortunately, the study all but ignores the impact on the lesions that define AD – plaques and tangles. Thus, the central message of "recapitulated pathophysiological features of AD in SARS-CoV-2 infection could readily be attributed to viral infection, alone. In addition, significant analytical details are missing as listed below which makes it difficult to interpret the finding. So, while important, this study requires further analysis and clarification before being published.Technical Concerns:1. The authors selected the significant gene changes based on only p-value and fold change. However, it is essential that the authors include the "false discovery rate (FDR)" values, which would greatly affect the validity and reliability of the findings. So, it is highly recommended that authors perform weighted correlation network analysis (WGCNA) to dissect minimally which gene modules and/or traits are affected in SARS, AD and AD with SARS-CoV-2 infected cases (https://doi.org/10.1186/1471-2105-9-559).

Regarding a network analysis, such as WGCNA, it is known that we should have approximately 15-20 samples to perform such an analysis. While the total number of samples available in this study does reach 16 samples, the samples are distributed across four different patient groups and two genders. Therefore, the mixture of groups and genders would make it difficult to identify condition-specific modules. The justification for using nominal p-values instead of FDR values was included in the Materials and methods (pages 19-20, lines 395-397) multiple testing correction was performed for all comparisons; however, due to the relatively small number of genes surpassing an adjusted p-value threshold of 0.05, a nominal p-value was used in the selection of differential genes for all downstream analyses. As this study is exploratory in nature, it is expected that the trends and pathways observed in the current study will provide direction for future investigations.

2. Figure 1 Gene expression profiles are well described but would be great if the authors could dissect up and down pathways based on the DEG changes. Figure 1 itself is not informative enough to get insights, so it is highly recommended that the authors report what genes/pathways are affected based on their observations.

It is important to note that the purpose of Figure1 is to facilitate quick visual identification of genes with large fold changes and statistical significance, as demonstrated through volcano plots and bar graphs, for subsequent analysis. As suggested by the reviewer, we revised the bar graphs in Figure 1 to indicate the number of genes that are upregulated or downregulated in each comparison (page 30, lines 594-596). Moreover, we want to point out that current analysis, depicted in Figure 3 and Figure 4, include the pathways that are either upregulated or downregulated based on the differential gene expression changes. These changes are detailed in the Figure 2C-source data 1 and Figure 2D-source data 2.

3. Figure 2 gene names are not described. Thus, it is hard to understand what is up or down based on the conditions.

We acknowledge this point. In Figure 2, we provided a comprehensive analysis of differentially expressed gene changes across all groups using a heatmap and included Rank-rank hypergeometric overlap analysis to further elucidate the findings in Figure 2—figure supplementary 1 and Figure 2—figure supplementary 2. To provide specific information regarding the changes, including upregulation and downregulation in the cortical BA9 and hippocampal formation, we provide detailed descriptions of the regulated pathways and genes in Figure 3 and Figure 4, respectively.

4. Figure 3 It is interesting that TREM1 signaling significantly changed in AD and AD with SARS-CoV-2, but more important that the authors should compare AD with SARS-CoV-2 vs AD condition, as well.

While this is an interesting direction, it should be noted that the scope of our current study is to explore the similarity of neuroinflammatory processes induced by SARS-CoV-2 infection to those observe in AD. Regarding TREM1 signaling, we now clarified it in the Results (page 8, lines 147-148) and Legend. Additionally, while a direct transcriptional comparison of AD with SARS-CoV-2 condition to the AD condition was examined (data not shown), the results were ambiguous and so not reported in this manuscript. This may be due to several factors, such as patient heterogeneity and potential differences in disease severity, and would require a larger sample size to reliably identify specific transcriptional modifications caused by SARS-CoV-2 infection in an AD background.

5. Figure 3 and Figure 4 Need to address/discuss why are the transcriptomic data from two regions – BA9 and HF – are showing different results.

We acknowledge the importance of understanding the regional distribution of transcriptomic changes and agree that it is a valuable aspect to consider. Our study was primarily designed to investigate the broader neuroinflammatory patterns induced by SARS-CoV-2 infection and its potential resemblance to that seen in AD. While regional variations in gene expression may exist, our research focused on identifying global transcriptomic changes, rather than region-specific alterations. This approach allowed us to capture the overall impact of SARS-CoV-2 infection on neuroinflammation across different brain regions, which provides a broad overview of the neuroinflammatory profile that will be explored further. Nonetheless, we recognize the significance of regional differences and their potential implications. Future studies with a specific focus on regional transcriptomic changes could provide valuable insights into the localized effects of SARS-CoV-2 infection on neuroinflammation.

6. Figure 5 The detailed experimental procedures are needed. e.g. is one dot representing the one individual subject? Or one dot is from the one image?

This is indeed an important question to consider. However, it should be noted that the specific subject we examined in our study (SARS-CoV-2) did not display amyloid plaque and tau-pathology. Moreover, as described in Table S1, the relatively short survival time of approximately 3 months post-infection may not be sufficient to observe significant changes in AD-related neuropathology such as amyloid-β plaques and Tau tangles. While we cannot provide a definitive answer to this question within the scope of our study, it is important to note that our findings strongly support the existence of significant similarities in proinflammatory conditions between SARS-CoV-2 and AD. This suggests the potential role of SARS-CoV-2 as a risk factor in triggering local proinflammatory changes, which are known to play a significant role in the onset of AD. Although the potential for SARS-CoV-2 infection promoting the development of AD is currently unknown, our findings reported herein support the notion that the long-term impact of infection on brain health contributes to and/or advances the progression to age-associated neurodegenerative disorders, including AD and AD-related dementias.

7. Figure 5. Microglial cells are less activated in brain parenchyma while they are accumulating more significantly around vessels. Thus, compromised blood vessels may be affected by activated microglial cells and consequently lead to more immune cell infiltration.

We would like to confirm that each point on the graphs represent the average finding per total brain area for an individual subject. We have amended the Materials and methods section (page 19, lines 379-382, and 385) and the Figure 5 and Figure 6 legends clarify this important point (page 32, lines 643-644 and page 33, lines 668-669).

8. As such, it would be useful if the authors could address whether they observed any peripheral immune cell infiltration such as T cells, B cells, neutrophils, and/or monocytes.

Leukocyte infiltrate can be identified by H&E, as well as through the hematoxylin counterstain used in the different IHC studies that examined specific proteins of interest. Naturally, when seen, leukocyte infiltration is obvious within the perivascular space, which was heavily viewed and analyzed on multiple 5µm sections on glass and through HALO analyses in silico. We would like to clarify that in our investigation, we did not observe evidence of leukocyte infiltration into the CNS compartment in any disease state examined. This is consistent with other reported findings from human autopsies and relevant animal models, which do not show leukocyte infiltration is a major finding in SARS-CoV-2 infection, outside of rare cases of encephalitis. While we can confirm that significant infiltrate was not seen in any of our subjects, this does not negate the potential for increased trafficking of leukocytes in the brain. We acknowledge that further studies will be necessary to determine the role of leukocytes in the observed pathophysiology. Future investigations should explore the potential involvement of peripheral immune cells, including T cells, B cells, neutrophils, and monocytes, in the context of SARS-CoV-2 infection and neuroinflammation.